# Maternal metabolic status and in-vitro culture conditions during embryonic genome activation deregulate the expression of energy-related genes in the bovine 16-cells embryo

**Maria B. Rabaglino[1], Niamh Forde[2], Urban Besenfelder[3], Vitezslav Havlicek[3], Helmut Blum[4], Alexander Graf[4], Eckhard Wolf[4], Patrick Lonergan[1] ***

**1** School of Agriculture and Food Science, University College Dublin, Dublin, Ireland, **2** Division of Reproduction and Early Development, School of Medicine, University of Leeds, Leeds, United Kingdom, **3** Reproduction Centre Wieselburg RCW, Institute for Animal Breeding and Genetics, University of Veterinary Medicine, Vienna, Austria, **4** Laboratory for Functional Genome Analysis, Gene Center, LMU, Munich, Germany

* pat.lonergan@ucd.ie

**Data Availability Statement:** Data were deposited in NCBI's Gene Expression Omnibus and are

## Abstract

The molecular consequences of the metabolic stress caused by milk production of dairy cows in the early embryo are largely unknown. The objective was to determine the impact of dam metabolic status or in vitro culture during embryonic genome activation (EGA) on the transcriptomic profiles of bovine 16-cell stage embryos. Two days after synchronized oestrus, in vitro produced 2- to 4-cell stage embryos were endoscopically transferred in pools of 50 into the oviduct ipsilateral to the corpus luteum of lactating (LACT, n = 3) or nonlactating (i.e. dried off immediately at calving; DRY, n = 3) dairy cows. On Day 4, the oviducts were flushed to recover the embryos. Pools of five Day-2 embryos (n = 5) and Day-4 16-cell stage embryos obtained in vitro (n = 3) or from LACT or DRY cows were subjected to RNAseq. Temporally differentially expressed genes (DEG; FDR<0.05) between Day-2 and Day-4 embryos were determined considering the differences between the three conditions under which EGA occurred. Also, DEG between Day-4 embryos derived from the three conditions were identified. Functional analysis of the temporal DEG demonstrated that genes involved in ribosome, translation and oxidative phosphorylation in the mitochondria were strongly more expressed in Day-4 than Day-2 embryos. Comparison of Day-4 embryos that underwent EGA in vitro, or in LACT or DRY cows, identified DEG enriching for mitochondrial respiration and protein translation, including the mTOR pathway. In conclusion, exposure of the embryo to an unfavourable maternal metabolic status during EGA influences its transcriptome and potentially the competence for pregnancy establishment.

accessible through GEO accession number GSE226844.

**Funding:** This research has received funding from the European Union Seventh Framework Programme FP7/2007-2013 under grant agreement n° 312097 ("FECUND"). MBR was funded by an H2020-MSCA-Individual Fellowship (Proposal 101021311). The funders had no role in study design, data collection and analysis, decision to publish, or preparation of the manuscript.

**Competing interests:** The authors have declared that no competing interests exist.

## Introduction

In mammals, several major developmental events occur during the first week of development following fertilization. These events include the first mitotic division, the timing of which has consequences for subsequent developmental competence [1], embryonic genome activation (EGA) [2], morula formation through the establishment of tight junctions between adjacent blastomeres in the developing embryo [3], blastocyst formation, involving the differentiation of inner cell mass and trophectoderm cells, and the onset of X-chromosome inactivation [4, 5]. Amongst these events, the switching on of the embryonic genome, which occurs at a species-specific stage, is arguably the most crucial for subsequent viability during development [6]. The EGA process occurs in two waves: a minor wave in the initial stage of active transcription followed by a second, major, wave, when widespread transcriptional activity of zygotic genes increases dramatically. This occurs between the 8- to 16-cell stage in bovine embryos [7, 8] and involves depletion of maternal transcripts stored in oocytes by degradation and translation, and their replacement by new embryo-specific transcripts. Indeed, many embryos fail to develop beyond this stage, due, at least in part, to insufficient expression of EGA-associated genes [9]. The environment in which the embryo develops in the first week after fertilization significantly affects blastocyst quality [10, 11]. In cattle, although the in vitro production (IVP) of embryos is now routine practice, the competence of the resulting embryos is often compromised [12]. Using alternative in vivo or in vitro culture (IVC) conditions before and during EGA, Gad, Hoelker [13] highlighted the critical influence of the environment in which EGA occurred on the blastocyst transcriptome. Culture conditions also affected the embryonic proteome around the time of EGA [14].

The physiological changes associated with milk production impact on circulating metabolites during the early postpartum period and have been implicated in poor reproductive efficiency in high-producing dairy cows [15, 16]. For example, the oviducts of postpartum lactating (LACT) cows, exposed to elevated concentrations of non-esterified fatty acids, beta-hydroxybutyrate (BHB), and reduced concentrations of insulin, IGF-I and glucose, were less capable of supporting development of early embryos to the blastocyst stage following endoscopic transfer of 2- to 4-cell embryos and recovery at Day 7, compared to the oviducts of heifers [17] or nonlactating (DRY) cows [16]. Thus, even if the embryo undergoes EGA in a more 'optimal' condition i.e., in an in vivo environment versus IVC, the dam status can alter the embryo transcriptomic profile and its developmental capacity. Therefore, we hypothesized that both IVC and the metabolic consequences of lactation would impact the transcriptome of the developing embryo around the time of genome activation.

Using a unique previously validated model [15, 16] involving drying off cows immediately after calving (to avoid lactation-induced metabolic stress) or milking them twice per day as is routine, the aims of this study were to characterize the effect of postpartum maternal metabolic status or culture in vitro on the embryonic transcriptome following EGA. We used high-throughput sequencing to generate comprehensive transcriptome profiles of Day 2 (pre-EGA) and Day 4 (post-EGA) bovine embryos following culture in vitro or in the oviducts of postpartum DRY or postpartum LACT dairy cows. Next, we identified clusters of differentially expressed genes (DEG) between the Day-4 16-cell embryo and the maternal mRNA in the Day-2 4-cell embryo, highlighting the differences between the three environmental conditions. Finally, we focused on the 16-cell stage to unravel the overall effects of the environment in which EGA occurred (in vitro or in vivo in the oviducts of dry or lactating cows) on the transcriptome.

## Materials and methods

### Animals

All experimental procedures involving animals were sanctioned by the Animal Research Ethics Committee of University College Dublin and were licensed by the Department of Health and Children, Ireland, in accordance with the Cruelty to Animals Act, 1876, and the European Community Directive 86/609/EC. The study was part of a larger study examining the influence of lactation-induced metabolic status on various aspects of the reproductive axis, including the follicular fluid metabolomic profile [15], the oviduct epithelial transcriptome [18], the oviduct fluid proteome [19], the uterine endometrial transcriptome [20] and the conceptus transcriptome and amino acid composition of uterine fluid [21]. The animal model used has been previously described in detail [15]. Briefly, 40 in-calf primiparous Holstein–Friesian heifers with a similar economic breeding index were enrolled into the study. At calving, cows were randomly assigned to one of two groups: (1) lactating (n = 20) or (2) non-lactating (n = 20). From calving, animals in the lactating group (LACT) were milked twice per day (07:00 and 16:00 h), while those in the non-lactating group were dried off immediately after calving (DRY; i.e. never milked). This model results in cows with very divergent metabolic status with LACT cows having higher serum concentrations of non-esterified fatty acids and beta-hydroxybutyrate, and lower glucose, insulin and insulin-like growth factor 1 (IGF1) compared with DRY cows [15, 16]. At approximately 50 days postpartum, oestrous cycles of DRY (n = 8) and LACT (n = 10) cows were synchronised by administration of a prostaglandin F2α analogue (PG; Estrumate, Intervet, Dublin, Ireland; equivalent to 0.5 mg Cloprostenol). Cows were observed for standing heat from 48 h following PG injection every 4 h (heat = Day 0). On the morning of Day 0 all animals received a 2.5 ml injection of Receptal (equivalent to 0.01 mg buserelin; Intervet, Dublin, Ireland) to ensure ovulation of the dominant follicle. On Day 2, IVP embryos were transferred in pools of 50 2- to 4-cell stage embryos into the oviduct ipsilateral to the CL via transvaginal endoscopic transfer as previously described [17]. On Day 4, the contents of the oviducts were flushed into the uterus using the same transvaginal endoscopic procedure and subsequently recovered by routine transcervical flushing of the uterus. The number and developmental stage of the structures recovered were recorded (Table 1). Embryos recovered from each individual cow at the 16-cell stage of development were snap frozen in pools of five in 1–2 μl of PBS and stored at -80˚C prior to RNA sequencing. The experimental design is illustrated in Fig 1.

### In vitro embryo production

Embryos were produced in vitro as previously described [11]. Briefly, immature cumulus–oocyte complexes (COCs) were recovered by aspirating follicles from the ovaries of heifers and cows slaughtered at a local abattoir, pooled, washed in PBS, and matured for 24 h in groups of 50 in 500 μL TCM-199 supplemented with 10% fetal calf serum and 10 ng/mL epidermal growth factor at 39˚C under an atmosphere of 5% $CO_2$ in air with maximum humidity. Matured COCs were inseminated with frozen-thawed sperm from one bull of proven fertility at a concentration of

**Table 1. Embryo recovery from postpartum nonlactating (DRY) or lactating (LACT) cows on Day 4 following oestrus.**

| Group | No. of structures recovered/No. transferred (%) | 16-cell No. structures (%) | 8- to 10-cell No. structures (%) | 4-cell No. structures (%) | Degenerate No. structures (%) |
|---|---|---|---|---|---|
| **DRY (n = 8)** | 252/400 (63.0%) | 95 (37.7%) | 58 (23.0%) | 22 (8.7%) | 80 (31.7%) |
| **LACT (n = 10)** | 275/500 (55%) | 102 (37.1%) | 68 (24.7%) | 46 (16.7%) | 58 (21.1%) |

All cows received 50 in vitro produced 2- to 4-cell stage embryos on Day 2 of the oestrous cycle via endoscopic oviduct transfer.

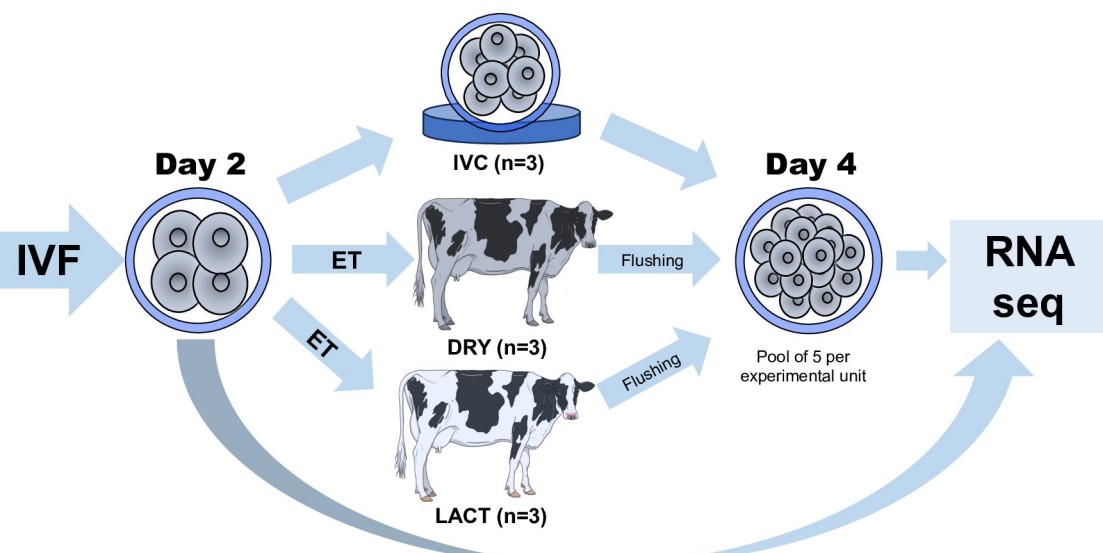

**Fig 1. Experimental design employed in this study.** Two days after synchronized oestrus, in vitro-produced embryos were transferred in pools of 50 Day-2 (IVF = Day 0) 2- to 4-cell stage embryos into the oviduct ipsilateral to the corpus luteum of postpartum dairy cows that were either milked (i.e. lactating) post-calving (LACT) or were dried off immediately at calving (i.e. nonlactating; DRY) via transvaginal endoscopic transfer. On Day 4, the oviducts were flushed and 16-cell stage embryos were snap-frozen in pools of 5 on an individual cow basis before mRNA extraction for RNAseq. A control group was maintained under in vitro culture (IVC) and processed in the same way for RNA Seq. Pools of five Day-2 embryos (n = 5) were also subjected to mRNA extraction for RNAseq.

$1 \times 10^6$ sperm/mL. Gametes were co-incubated for 20 h at 39°C in an atmosphere of 5% $CO_2$ in air with maximum humidity. Presumptive zygotes were denuded by gentle vortexing and cultured in synthetic oviduct fluid droplets supplemented with 3 mg/ml of BSA (25 μL; 25 embryos per droplet) at 39°C in a humidified atmosphere with 5% $CO_2$ and 5% $O_2$ under mineral oil.

## RNA sequencing of embryos

RNA extraction was carried out on pools of five Day-2 embryos (n = 5) or pools of five Day-4 embryos following culture in vitro (IVP; n = 3) or recovery from DRY (n = 3) or LACT (n = 3) cows. For the in vivo conditions, the five embryos in each pool were collected from the same cow. Extracted RNA was used to construct sequencing libraries using the Encore Complete RNASeq library system of NuGEN. A minimum of 100 ng of total RNA was used and samples were enriched for non- rRNA during cDNA synthesis. No ribosomal depletion or polyA-RNA enrichment steps were carried out while all non-rRNA were captured. This also allowed us to retain RNA strand information. All libraries were sequenced on the Illumina HiSeq 1500 as 100 bp single end reads. After demultiplexing, adaptor sequences and poly A tails were removed and an additional number of sequencing runs was performed to ensure sufficient coverage, generating ~45 million raw reads per sample in FASQ format.

## Bioinformatic analyses

The read qualities for each FASTQ file were accessed with FastQC https://www.bioinformatics.babraham.ac.uk/projects/fastqc/), and low-quality bases and adapters were removed with Trimmomatic (V 0.39) [22]. The sequenced reads were mapped to the bovine reference genome (bosTau 9) with the STAR aligner (V 2.7.0b) [23]. On average, 81.7% of the reads were uniquely mapped to the genome, ranging from 71% to 86.8%. Read counts were

estimated at the gene level, and the counting was done with featureCounts [24], which is part of the Subread software (V 2.0.3). Relatedness of all samples according to the whole transcriptome was assessed with a principal component analysis (PCA) using internal packages of R. Read counts were normalized through variance stabilizing transformation implemented with the DESeq2 package [25] before running the PCA.

Data were deposited in NCBI's Gene Expression Omnibus and are accessible through GEO accession number GSE226844.

### Differential expression analysis

Differences between Day-2 and Day-4 embryos which underwent EGA in three different conditions, were estimated with the maSigPro package [26]. Briefly, this method fits a regression model to the data, modelled by a negative binomial distribution, to identify DEG between time points, corrected by false discovery rate (FDR) at FDR<0.05. Next, DEG between experimental groups are identified through a stepwise regression. Coefficients derived from this regression step were employed to cluster DEG with similar expression patterns.

Differences between the three groups only for Day-4 embryos were determined with the edgeR package [27]. Genes with less than one count per million in three or more samples (smaller class) were filtered out before normalization [28]. Filtered data was normalized through weighted trimmed mean of M-values [29]. Next, observation weights were used for robust estimate of the negative binomial dispersion parameter for each gene and for estimating regression parameters. Finally, a negative binomial generalized log-linear model was fit to read counts for each transcript and conduct genewise likelihood ratio tests for the coefficient contrast [30]. The matrix of contrast was built based on the comparisons between the groups. Differentially expressed genes were defined as those with an FDR<0.05. Functional analysis of the DEG was carried out using Database for Annotation, Visualization and Integrated Discovery (DAVID; [31], to determine enriched terms (FDR<0.05).

Identified DEG between Day-2 and Day-4 embryos, and between groups in Day-4 embryos, were subjected to hierarchical clustering according to their expression profile, using Spearman Rank Correlation as similarity metric and centroid linkage as clustering method, implemented with the Cluster 3.0 software [32]. The resulting dendrogram and the heat map were visualized with Java TreeView [33].

Results from the differential expression analysis were explored to identify enriched ontological terms of biological relevance, and the expressions of the related DEG for each sample across all conditions were depicted in line plots. Data were normalized through variance stabilizing transformation [34] and standardized before plotting. Finally, relevant ontological terms de-regulated between IVC, DRY and LACT conditions were employed for network analysis with the Cytoscape software [35]. The entire list of genes involved in the enriched term was downloaded from the corresponding database (KEGG pahway or AmiGO 2) and a unique network was inferred using the GeneMania plugin for Cytoscape [36]. The topological parameters of the resulting network were estimated with NetwokAnalyzer [37]. Next, the nodes corresponding to DEG enriching the ontological terms were isolated and colored according to the normalized expression in Day-2 embryos and each condition of Day-4 embryos.

## Results

### Differences between the Day-2 and Day-4 embryonic transcriptome

The PCA plot demonstrates a strong separation between the transcriptomes of Day-2 and Day-4 embryos. Additionally, for Day-4 embryos, the transcriptomes of LACT and IVC were more similar to each other than to that of DRY embryos (S1 Fig).

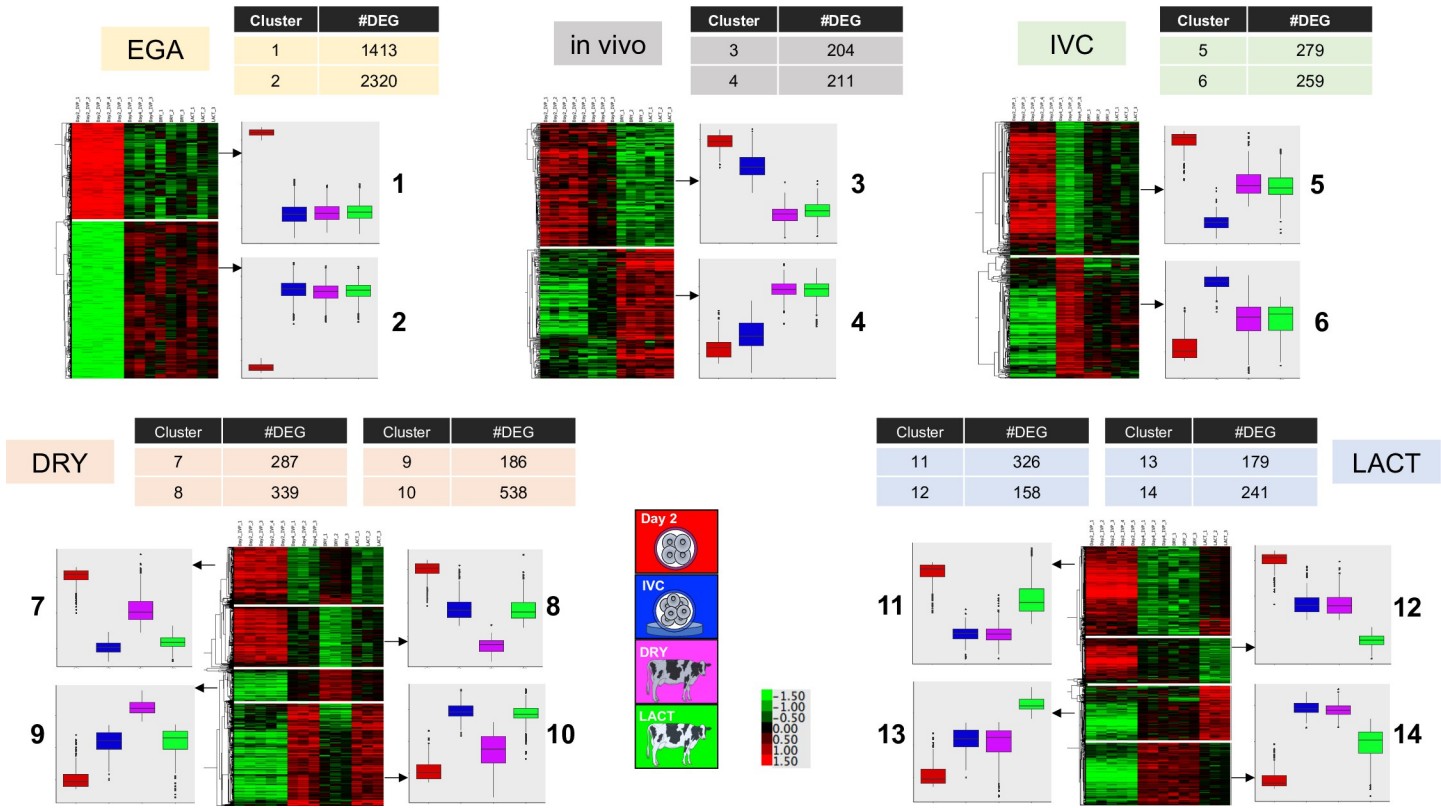

**Fig 2. Differentially expressed genes (DEG) between Day-2 and Day-4 embryos undergoing embryonic genome activation in three environmental conditions.** The identified DEG were organized into five groups according to the condition that was driving the gene expression. Additionally, DEG in each group could be separated into clusters according to the direction of the gene expression. The boxplots represent the average expression for genes in each cluster for each condition. IVC: in vitro culture; DRY: nonlactating dairy cows, LACT: lactating dairy cows.

There were 6940 DEG between Day-2 and Day-4 embryos and among the three conditions on Day 4. These DEG were classified into five groups according to the condition driving the gene expression profile (Fig 2). The groups were:

- DEG centrally involved in EGA: genes changing in expression from Day 2 to Day 4, but not differentially expressed between the Day-4 groups (IVP, DRY and LACT).

- DEG affected by in vivo conditions: genes changing in expression at Day 4 in both DRY and LACT embryos compared to Day-2 and IVC embryos.

- DEG affected by IVC: genes changing in expression at Day 4 in IVC embryos compared to Day-2 and both DRY and LACT embryos.

- DEG influenced by the maternal metabolic status (DRY or LACT): genes changing in expression at Day 4 in DRY embryos compared to Day 2 and IVC and LACT embryos, or in LACT embryos compared to Day-2 and IVC and DRY embryos.

Additionally, DEG in each group could be separated into two clusters (for DEG in "EGA", "in vivo" and "IVC" groups) or four clusters (for DEG in "DRY" and "LACT" groups) according to the direction of the gene expression, making 14 clusters in total. Fig 2 summarizes the number of genes per cluster in each group and depicts the gene direction in heat maps and box plots. Associated genes are listed in S1 Table.

Functional analysis of DEG in each cluster revealed that 1413 Day 2 genes (i.e., maternally expressed), that were not expressed by Day-4 embryos (Cluster 1), enriched ontological terms related to cell cycle, while those 2320 genes expressed by Day-4 embryos, but not at Day 2 (Cluster 2), were strongly involved in ribosome, translation and oxidative phosphorylation in the mitochondria. The 211 genes more expressed by Day-4 embryos that underwent EGA in vivo compared to in vitro (Cluster 4) enriched terms related to epigenetic modifications, such as methylation, histones, and chromatin silencing. On the other hand, several of the 259 genes more expressed by Day-4 embryos that underwent EGA during IVC compared to in vivo (Cluster 6) were involved in formation of extracellular vesicles (exosomes) and endosomes. Regarding genes affected by the maternal metabolic status, embryos that underwent EGA in DRY exhibited higher expression of 287 genes than in IVC and LACT embryos, although with lower expression levels than the maternal mRNA at Day 2 (Cluster 7); these genes enriched for epigenetic modification terms, such as methylation and histones, and DNA repair. In addition, 538 genes less expressed than in IVC and LACT embryos, but with higher expression levels than the maternal mRNA at Day 2 (Cluster 10), enriched terms involved in protein synthesis (ribosome and translation). Finally, embryos that underwent EGA in LACT showed 241 genes less expressed than in IVC and DRY embryos, although more expressed than in Day-2 embryos (Cluster 14). These genes enriched terms related to energy regulation, such as oxido-reductase and TCA cycle. The full lists of enriched terms for each cluster are detailed in S2 Table.

## Differences between maternal conditions and in vitro culture in the Day 4 embryos transcriptome

Comparisons between the transcriptomes of Day 4 embryos only (without considering Day 2 embryos) revealed 3451 DEG between the three conditions (DRY, LACT, and IVC) that were distributed in six clusters, named A to F (Fig 3 and S3 Table). Genes in Clusters A and B (more

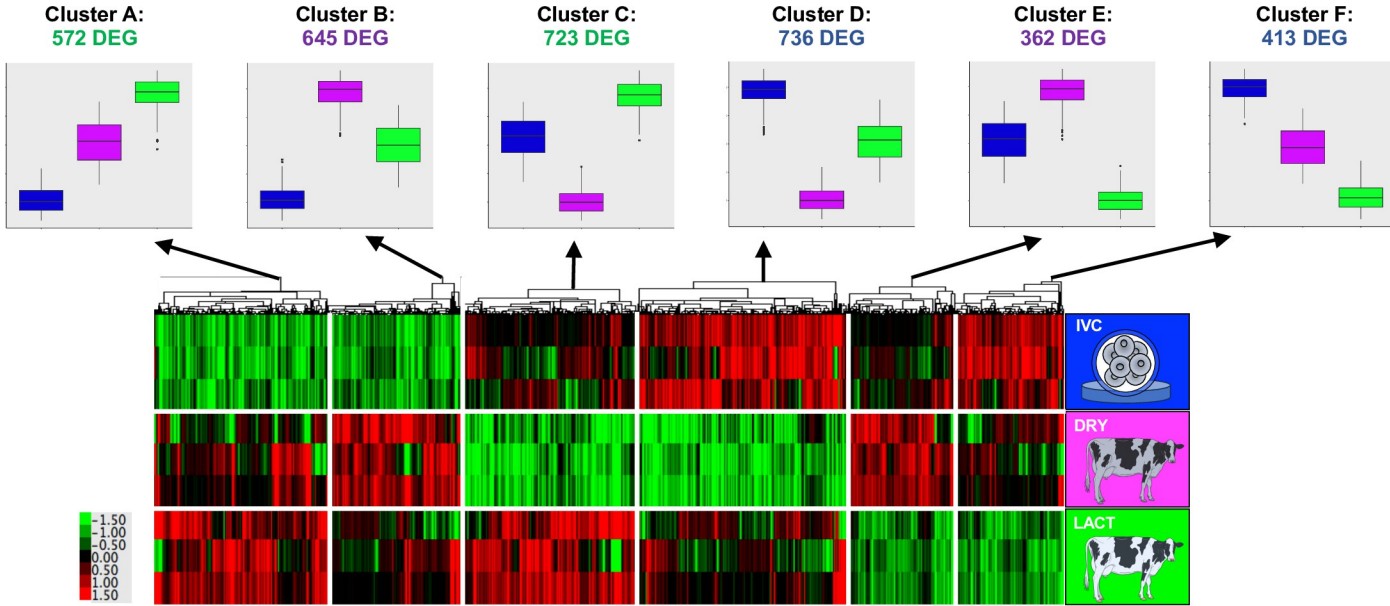

**Fig 3. Differentially expressed genes (DEG) in Day-4 embryos undergoing embryonic genome activation in three environmental conditions.** The identified DEG were classified into six clusters according to the direction of the gene expression. The boxplots represent the average expression for genes in each cluster for each condition. IVC: in vitro culture; DRY: nonlactating dairy cows, LACT: lactating dairy cows.

expressed in the in vivo condition than in IVC) enriched terms related to epigenetic modifications. In addition, the 645 DEG more expressed in DRY than LACT and IVC (Cluster B) also enriched for the mechanistic target of rapamycin kinase (mTOR) signalling pathway. Several of the 736 DEG more expressed in IVC, followed by LACT and DRY (Cluster D) were involved in protein synthesis terms, specifically translation initiation, while the 413 DEG, also highly expressed in IVC but followed by DRY and LACT (Cluster F), enriched for exosomes and endosomes. Finally, the 362 DEG strongly expressed in DRY, followed by IVC, and with low expression levels in LACT (Cluster E), were enriching terms related to energy generation (oxidative phosphorylation and TCA cycle). All the enriched terms by genes in each cluster are listed in S4 Table. For all clusters, DEG identified by both analyses (considering or not the Day-2 embryos) enriched, as expected, terms related to the cellular regions such as nucleus, nucleoplasm, nucleolus, cytoplasm, cytosol, etc.

### Exploration of selected ontological terms

Results from the functional analysis described above suggested that processes related to mitochondrial function and protein synthesis occur in Day-4 but not in Day-2 embryos. In other words, genes involved in these processes are being expressed by the embryonic genome after activation but not by the maternal mRNA. However, expression levels for several of these genes depend on the environment where EGA takes place. That is, some genes involved in oxidative phosphorylation were more expressed when EGA occurred in DRY than in IVC, and less expressed in LACT. On the other hand, genes involved in translation initiation, in particular in the formation of the 48S and 43S preinitiation complex, were more expressed if EGA occurred in IVC than in DRY or LACT. Interestingly, several genes involved in the mTOR pathway, a key regulator of energy-sensing pathways, were more expressed in embryos that underwent EGA in DRY compared to LACT, and less expressed in IVC. Fig 4 depicts the expression levels for genes involved in the aforementioned ontological terms. This figure also shows the expression levels for genes enriching for methylation and chromatin silencing, which were more expressed in Day-4 embryos undergoing EGA in vivo, and for genes involved in vesicle formation, which had higher expression in embryos cultured in vitro than in the other conditions.

### Network analysis of selected ontological terms

Given the crucial role of energy regulatory mechanisms in embryo development, all the genes (in addition to the DEG) involved in the mTOR pathway, oxidative phosphorylation, and translation initiation were subjected to network analysis. The whole inferred network exhibited, as expected, high connectivity among all of the genes, including the DEG, involved in these processes (S2 Fig).

Topological parameters for the network indicated that the average degrees (related to the number of edges linked to a node) for all genes in the mTOR pathway, oxidative phosphorylation, and translation initiation were 73.5, 177, and 139. The average degrees for DEG on each pathway were 86.1, 192, and 163.3, evidencing high connectivity for these DEG with the rest of the genes in the pathway. Fig 5 shows the contrasts in expression levels of these DEG for Day-2 embryos and Day-4 embryos undergoing EGA in different environments, while Table 2 specifies the fold change (FC) for each gene between the three conditions for Day-4 embryos. Briefly, if the embryo undergoes EGA in a favourable environment (DRY in this study) certain genes in the mTOR pathway are more expressed, potentially de-regulating the other genes in the pathway, leading to up-regulation of key genes in the oxidative phosphorylation pathway and possibly to the whole pathway (S3 Fig), and down-regulating genes involved in translation

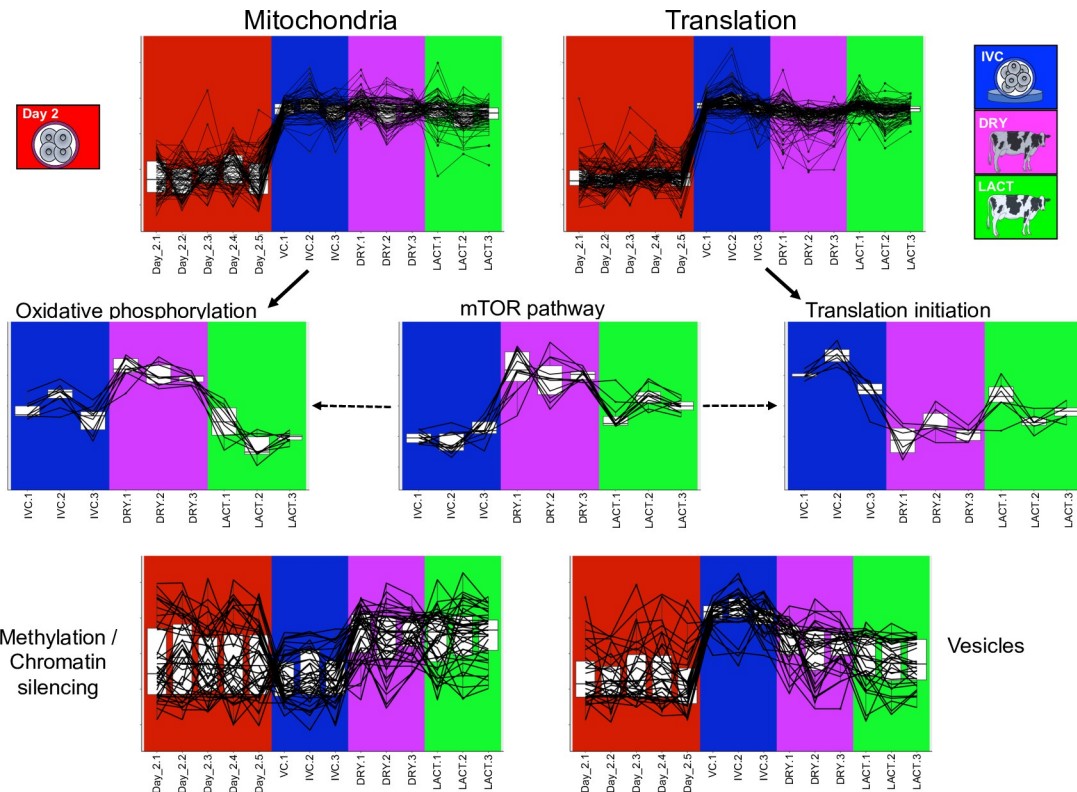

**Fig 4. Expression levels for genes involved in selected enriched ontological terms by the differentially expressed genes.** Gene expression was measured in Day-2 embryos or Day-4 embryos undergoing embryonic genome activation (EGA) in one of three conditions (IVC, DRY, or LACT). IVC: in vitro culture; DRY: nonlactating dairy cows, LACT: lactating dairy cows.

initiation. These genes, part of the maternal mTOR pathway, are not up-regulated in more stressful conditions for the embryo, such as in IVC or in a cow with an adverse metabolic status (LACT).

## Discussion

The window of time during which EGA occurs is one of the most critical periods during early embryogenesis, since a proper activation of the gene expression machinery determines the developmental fate of the embryo [38, 39]. Thus, the environment in which the embryo undergoes EGA can influence this phenomenon and potentially have long-term consequences for the embryo and later for the foetus. Several studies have characterized the transcriptomic events occurring during the period of development from the oocyte to the blastocyst, including to the 8- to 16-cell stage, that is, covering the period of EGA [40–44]. Others have compared the effect of IVC at different stages on the embryonic transcriptome [13] and methylome [45]. Here, we have expanded our knowledge of this key embryonic event by comparing the transcriptome of 16-cell embryos undergoing EGA not only in in vitro vs in vivo but also, characterizing the impact of lactation on this key event. While the impact of IVP on the embryo transcriptome during this period has been described, as mentioned below, the potential effects of the maternal metabolic status on the process are less understood. In a previous study [16], approximately 65 Day-2 IVP embryos (2- to 4-cell stage) were endoscopically transferred to the oviducts of 60 days post-partum dry or lactating dairy cows two days after being observed in heat. Systematic blood collection from 15 d before calving to approximately 100 d

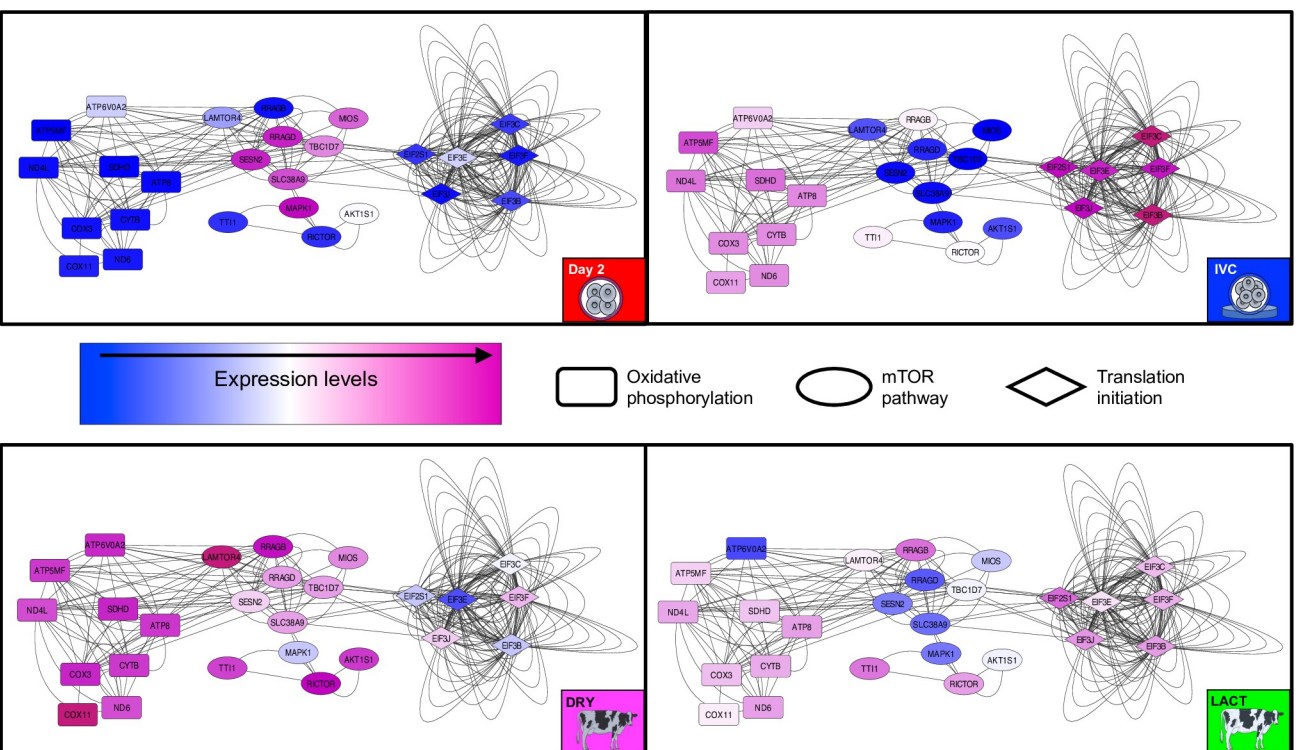

**Fig 5. Network representation for differentially expressed genes enriching ontological terms involved in energy regulation.** The nodes are coloured according to the expression levels on Day-2 embryos or Day-4 embryos undergoing embryonic genome activation (EGA) in one of three conditions (IVC, DRY, or LACT). IVC: in vitro culture; DRY: nonlactating dairy cows, LACT: lactating dairy cows.

postpartum confirmed the altered metabolism of lactating cows, as they had divergent concentrations of non-esterified fatty acids, β-hydroxybutyrate, glucose, insulin, and insulin-like growth factor-I. Fewer embryos had developed to the blastocyst stage in the lactating cows compared with the dry cows (33% vs 49%, respectively) when recovered five days after transfer. Interestingly, no differences were observed in the Day-15 conceptus developmental rate when Day-7 blastocysts were transferred to the same animals 30 days later [16]. Thus, the oviductal environment of lactating cows was adverse for early embryo development before the blastocyst stage when compared to dry cows. Here, we complement and reinforce this previous study by characterizing the molecular consequences of the metabolic status induced by lactation on the early embryo immediately after undergoing EGA. A limitation of this study resides in the sample size per group (n = 3) for the Day-4 embryos. Each experimental unit was constituted by a pool of five embryos and thus, embryos that underwent EGA in the oviduct of the same DRY or LACT cow would exhibit less variability. However, a PCA of the whole transcriptome showed that samples from each group clustered together in the plot, supporting the treatment effect on the embryo transcriptome.

Results show that genes that could be influencing the epigenome, such as those encoding for histones H2A, H3, and H4 and the methylation process, exhibited similar expression between IVP embryos at the 2- to 4-cell (pre-EGA) and 8 to 16-cell (post-EGA) stage but switched to increased expression in the Day-4 embryos that underwent EGA in vivo, with no differences between DRY and LACT. Genes involved in epigenetic modifications are present in both maternal and embryonic transcripts, as determined by analysing the transcriptome of 8- to 16-cell embryos that developed in the presence or absence of alpha-amanitin, an inhibitor

**Table 2. Differentially expressed genes among embryos that underwent embryonic gene activation in different conditions enriching for ontological terms involved in energy regulation.**

| Ontological term | Ensembl ID | Name | Symbol | FC DRY vs LACT | FC LACT vs IVC | FC DRY vs IVC |
|---|---|---|---|---|---|---|
| Oxidative Phosphorylation | ENSBTAG00000002094 | ATP synthase membrane subunit f | ATP5MF | 2.55 | -2.24 | 1.13 |
| | ENSBTAG00000007015 | cytochrome c oxidase copper chaperone COX11 | COX11 | 3.30 | -1.47 | 2.24 |
| | ENSBTAG00000007272 | ATPase H+ transporting V0 subunit a2 | ATP6V0A2 | 2.66 | -1.74 | 1.53 |
| | ENSBTAG00000016266 | succinate dehydrogenase complex subunit D | SDHD | 2.45 | -1.59 | 1.54 |
| | ENSBTAG00000043546 | NADH dehydrogenase subunit 6 | ND6 | 1.48 | -1.08 | 1.37 |
| | ENSBTAG00000043550 | cytochrome b | CYTB | 1.58 | -1.09 | 1.45 |
| | ENSBTAG00000043559 | NADH dehydrogenase subunit 4L | ND4L | 1.47 | -1.20 | 1.22 |
| | ENSBTAG00000043560 | cytochrome c oxidase subunit III | COX3 | 1.69 | -1.15 | 1.48 |
| | ENSBTAG00000043564 | ATP synthase F0 subunit 8 | ATP8 | 1.64 | -1.08 | 1.52 |
| mTOR pathway | ENSBTAG00000002363 | sestrin 2 | SESN2 | 2.72 | 3.29 | 8.95 |
| | ENSBTAG00000004100 | TELO2 interacting protein 1 | TTI1 | 1.18 | 1.65 | 1.94 |
| | ENSBTAG00000005763 | AKT1 substrate 1 | AKT1S1 | 2.09 | 2.21 | 4.62 |
| | ENSBTAG00000006697 | RPTOR independent companion of MTOR complex 2 | RICTOR | 1.27 | 1.25 | 1.58 |
| | ENSBTAG00000010312 | mitogen-activated protein kinase 1 | MAPK1 | 1.35 | 1.53 | 2.06 |
| | ENSBTAG00000012980 | Ras related GTP binding B | RRAGB | 2.53 | 4.45 | 11.25 |
| | ENSBTAG00000013781 | meiosis regulator for oocyte development | MIOS | 1.30 | 1.99 | 2.58 |
| | ENSBTAG00000015009 | late endosomal/lysosomal adaptor, MAPK and MTOR activator 4 | LAMTOR4 | 19.95 | 181.95 | 3629.44 |
| | ENSBTAG00000017160 | TBC1 domain family member 7 | TBC1D7 | 1.37 | 4.07 | 5.58 |
| | ENSBTAG00000018438 | Ras related GTP binding D | RRAGD | 4.17 | 1.68 | 7.01 |
| | ENSBTAG00000033313 | solute carrier family 38 member 9 | SLC38A9 | 2.14 | 1.53 | 3.27 |
| Translation initiation | ENSBTAG00000000359 | eukaryotic translation initiation factor 3 subunit J | EIF3J | -1.21 | -1.65 | -2.01 |
| | ENSBTAG00000004861 | eukaryotic translation initiation factor 3 subunit F | EIF3F | -1.07 | -1.93 | -2.07 |
| | ENSBTAG00000006543 | eukaryotic translation initiation factor 3, subunit C-like | EIF3CL | -1.24 | -1.69 | -2.09 |
| | ENSBTAG00000006702 | eukaryotic translation initiation factor 3 subunit E | EIF3E | -1.39 | -1.27 | -1.77 |
| | ENSBTAG00000007474 | eukaryotic translation initiation factor 3 subunit B | EIF3B | -1.45 | -1.46 | -2.12 |
| | ENSBTAG00000016311 | eukaryotic translation initiation factor 2 subunit alpha | EIF2S1 | -1.61 | -1.17 | -1.89 |

IVC: in vitro culture; DRY: nonlactating dairy cows, LACT: lactating dairy cows.

FC: fold change.

of eukaryotic RNA polymerase [41, 42]. While we cannot know if these differences in the transcriptome were associated with changes in DNA methylation levels in the embryos from this study, a previous report showed that embryos produced in vivo but subjected to IVC before or during EGA exhibited strong changes in DNA methylation [46].

Another result from our study demonstrates the strong expression of genes related to mitochondrial function and translation in Day-4 compared to Day-2 embryos. Accordingly, other authors have identified genes enriching for oxidative phosphorylation and citrate cycle [40, 44] and translation [42–44] in Day-4 embryos compared to earlier stages. Nonetheless, the experimental design employed in this study allowed us to discern how these vital biological processes might be triggered in the embryo according to the environment undergoing EGA. Clearly, certain genes involved in oxidative phosphorylation were up-regulated in DRY compared to IVC and LACT (average FC: 1.50 and 2.09, respectively). Interestingly, several genes

involved in the mTOR pathway, a key regulator of energy sensing in the cell, were also strongly more expressed in DRY than LACT (average FC: 3.64) and even more when compared to IVC (average FC: 334.40). In contrast, six of the 17 genes compromising the eukaryotic 48S preinitiation complex were down-regulated in DRY compared to both LACT and IVC (average FC: -1.33 and -2.00, respectively).

One of the up-regulated genes in the mTOR pathway was the solute carrier family 38 member 9 (*SLC38A9*), which binds and transports amino acids in the lysosome and controls mTORC1 activity in response to amino acids [47]. The protein encoded by this gene interacts with the Ragulator subunit late endosomal/lysosomal adaptor, MAPK and MTOR activator 4 (*LAMTOR4*), which was markedly up-regulated in DRY compared to IVC (more than 3000 FC). The Ragulator complex regulates the Ras related GTP binding (*RRAG*) proteins, which form a heterodimeric complex between *RRAGB* and *RRAGD*, promoting *MTORC1* recruitment to the lysosome and activation [48]. Both *RRAGB* and *RRAGD* were strongly up-regulated in DRY compared to IVC (FC: 11.25 and 7.01, respectively). Paradoxically, the gene encoding for AKT1 substrate 1 (*AKT1S1*) was also upregulated, which binds the regulatory-associated protein of mTOR (raptor) and suppresses mTORC1 [49], while the gene encoding for mTOR was not de-regulated. This is not surprising as the sequencing technique takes a snapshot of a specific time point in the cell, and activation of the mTORC1 complex could have occurred at another time, or the sample size limited identifying deregulated expression of this gene. It has been shown that *MTORC1* stimulates the expression of genes in all the mitochondrial complexes, inducing oxidative phosphorylation in human trophoblastic cells [50]. Here, key genes in all the complexes of the respiratory chain (I through V) were strongly up-regulated in DRY when compared to LACT, which in turn were downregulated when compared to IVC. Finally, another vital function of the mTOR signalling pathway is to promote the assembly of the eukaryotic translation initiation factor 4F (*eIF4F*) complex [51] which interacts with the large multi-subunit protein eIF3 within the 43S pre-initiation complex for cap-dependent translation initiation [52]. Genes encoding for these proteins were lowly expressed in DRY compared to both IVC and LACT which in turn were also downregulated when compared to IVC.

These main molecular changes observed between the three Day-4 embryo groups reflect the adaption mechanisms carried out by the embryo in response to the environment. Embryos that underwent EGA in the oviduct of LACT cows experienced lower energetic availability, as these animals were under negative energy balance, manifested by lower circulating concentrations of glucose and higher concentrations of BHB [15]. The low glucose environment can lead to depletion of ATP and an increase in AMP concentration, which activates the AMPK and MAPK kinases, resulting in an inhibition of the mTOR pathway [53, 54]. As mentioned above, the mTOR pathway regulates mitochondrial function (oxidative phosphorylation) and translation initiation. This latter mechanism takes place under conditions of cell stress to induce the translation of specific mRNA and diminish the cell damage or induce apoptosis otherwise [55]. Therefore, the "nutrient-poor" condition faced by the Day-2 embryos in the LACT cows resulted in an inhibition of the mTOR pathway and activation of translation initiation as early as the 16-cell stage when compared to Day-4 embryos collected from the DRY cows. These modifications can persist in later stages. Indeed, de-regulation of the mTOR pathway, including oxidative phosphorylation and protein translation, was also observed in blastocysts obtained from dairy cows with high circulating concentrations of beta-hydroxybutyrate at 60 days postpartum when compared to embryos obtained from cows with low concentrations [56]. This study also demonstrated altered methylation patterns consistent with the transcriptomic results.

Similarities between the transcriptomic profile of LACT and IVP embryos might arise from the fact that embryos are exposed to suboptimal conditions in both scenarios, forcing the embryo to adopt a metabolic signature described as "economy mode", characterized by mitochondrial disfunction [57]. The in vitro process is still associated with a suboptimal stressful environment for the embryo, despite all the advancements done in this area, which impact both the embryo epigenome and transcriptome, as described by several investigators [11, 58–62]. Most of these studies demonstrated that the main consequence of the in vitro process is de-regulation of the mTOR pathway and changes in the DNA methylation levels of energy-sensing genes, potentially leading to permanent consequences. Indeed, genes involved in energy homeostasis were affected in the muscle and liver of IVP postnatal dairy calves when compared to calves derived from and in vivo embryo [63]. Given the similarities between bovine and human embryos in regulatory mechanisms and developmental transcriptomic dynamics [64], results from the present study can also have implications for women's reproductive health.

In conclusion, results from this study demonstrate that EGA involves the expression of genes involved in mitochondrial function and translation, as previously observed by other authors. Furthermore, the environment in which EGA takes place strongly influences these mechanisms, likely through regulation of the mTOR signalling pathway. The expression of key genes involved in this pathway are stimulated in the embryo if EGA occurs in the oviduct but not when EGA occurs in vitro. Furthermore, for the in vivo conditions, the level of gene expression is strongly influenced by the dam's metabolic status, with the most robust expression occurring in the most optimal condition, i.e., in DRY cows compared to LACT cows in this study. Deregulation of this critical energy-regulatory mechanism affects, as a consequence, other essential cellular processes in the embryo, such as oxidative phosphorylation and protein translation.

## Supporting information

**S1 Checklist. PLOS ONE clinical studies checklist.**
(DOCX)

**S1 Fig. Principal component analysis plot of the Day-2 and Day-4 embryo transcriptome.** Day 4 embryos underwent embryonic genome activation during in-vitro culture (IVC) or in the oviduct of nonlactating (DRY) or lactating (LACT) dairy cows.
(TIF)

**S2 Fig. Networks constructed with all the genes involved in the selected pathways.** The highlighted nodes are genes differentially expressed between Day-4 embryos that underwent embryonic genome activation in different conditions.
(TIF)

**S3 Fig. Pathway maps drawn by the Kyoto encyclopaedia of genes and genomes (KEGG).** Genes in purple are differentially expressed genes between Day-4 embryos that underwent embryonic genome activation in different conditions.
(TIF)

**S1 Table. Differentially expressed genes per group and cluster.** Differentially expressed genes were determined between Day-2 and Day-4 embryos that underwent embryonic genome activation under different conditions (see Fig 2).
(XLSX)

**S2 Table. Enriched ontological terms with the genes in each cluster.** Genes in each cluster are listed in S1 Table.
(XLSX)

**S3 Table. Differentially expressed genes per cluster.** Differentially expressed genes were determined between Day-4 embryos that underwent embryonic genome activation under different conditions (see Fig 3).
(XLSX)

**S4 Table. Enriched ontological terms with the genes in each cluster.** Genes in each cluster are listed in S3 Table.
(XLSX)

## Author Contributions

**Conceptualization:** Patrick Lonergan.

**Data curation:** Maria B. Rabaglino, Helmut Blum, Alexander Graf.

**Formal analysis:** Maria B. Rabaglino, Helmut Blum, Alexander Graf.

**Funding acquisition:** Urban Besenfelder, Eckhard Wolf, Patrick Lonergan.

**Investigation:** Niamh Forde, Urban Besenfelder, Vitezslav Havlicek, Patrick Lonergan.

**Methodology:** Maria B. Rabaglino, Niamh Forde, Urban Besenfelder, Vitezslav Havlicek, Helmut Blum, Patrick Lonergan.

**Project administration:** Eckhard Wolf, Patrick Lonergan.

**Supervision:** Patrick Lonergan.

**Visualization:** Maria B. Rabaglino.

**Writing – original draft:** Maria B. Rabaglino.

**Writing – review & editing:** Maria B. Rabaglino, Niamh Forde, Urban Besenfelder, Vitezslav Havlicek, Helmut Blum, Alexander Graf, Eckhard Wolf, Patrick Lonergan.

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
