## [Decision Letter · Decision Letter 0]

7 Jul 2023

PONE-D-23-15571Maternal metabolic status and in-vitro culture conditions during embryonic genome activation deregulate the expression of energy-related genes in the bovine 16-cells embryoPLOS ONE

Dear Dr. Lonergan,

,

Thank you for submitting your manuscript to PLOS ONE. After careful consideration, we feel that it has merit but does not fully meet PLOS ONE’s publication criteria as it currently stands. Therefore, we invite you to submit a revised version of the manuscript that addresses the points raised during the review process.

ACADEMIC EDITOR:  Although both reviewers found some interesting findings, they also have major concerns. Please respond to the reviewers' questions and submit the updated version of the manuscript. Particularly, the discussion should be made more precisely based on the findings of this study.

We look forward to receiving your revised manuscript.

Kind regards,

Birendra Mishra, DVM, PhD

Academic Editor

PLOS ONE

3. We note that Figure 1 in your submission contain copyrighted images. All PLOS content is published under the Creative Commons Attribution License (CC BY 4.0), which means that the manuscript, images, and Supporting Information files will be freely available online, and any third party is permitted to access, download, copy, distribute, and use these materials in any way, even commercially, with proper attribution. For more information, see our copyright guidelines: http://journals.plos.org/plosone/s/licenses-and-copyright.

Reviewers' comments:

Reviewer's Responses to Questions

**Comments to the Author**

1. Is the manuscript technically sound, and do the data support the conclusions?

Reviewer #1: Partly

Reviewer #2: Yes

2. Has the statistical analysis been performed appropriately and rigorously? 

Reviewer #1: Yes

Reviewer #2: Yes

3. Have the authors made all data underlying the findings in their manuscript fully available?

Reviewer #1: Yes

Reviewer #2: Yes

4. Is the manuscript presented in an intelligible fashion and written in standard English?

Reviewer #1: Yes

Reviewer #2: Yes

5. Review Comments to the Author

Reviewer #1: This study explores the impact of metabolic stress caused by milk production of dairy cows in the early embryo, in particular during the -major- embryonic genome activation (EGA). The transcriptome of embryos entirely cultured in vitro was examined in contrast with embryos fertilized in vitro that developed in the genital tract of dry and lactating cows. The analysis identified a number of (new) genes / pathways differentially expressed between all types of embryos used. The design of this study is elegant, the work is well written and easy to read.

The main limitations of this study are the following

- High dimensionality: in spite the correct limitative statistical tests for false discovery, authors should be cautious when drawing conclusions from this manuscript. Variability is also limited within in vivo embryos, as all them were collected from a same cow (L159-160). The above concerns must be discussed

- Limited sample numbers: The use of N=3 groups per treatment in large data experiments is discouraged, as detection of outliers is more difficult and, in the case of strong divergences between one and the other two experimental groups, discarding one group ruins the experiment.

- Some inconsistencies of the Discussion section need re-thinking:

L386 onwards: the results dealing with epigenetic control (i.e., H2A, H3, and H4 and the methylation process, exhibited similar expression between IVP embryos at the 2- to 4-cell (pre-EGA) and 8 to 16-cell (post-EGA)) can be difficult to explain. The authors provide appropriate references to EGA activation (42 and 43), but the facts are not the same. In contrast, other results obtained in this study appear more consistent (L399-400)

L423-425: Or simply is an inconsistent result obtained from sample limitations

Other concerns

L39-41 (abstract): unclear sentence that should be re-rewritten.

L99-L112: Consider whether the text described is relevant for this study.

L-244-245: re-write as follows: Functional analysis of DEG in each cluster revealed that 1413 Day 2 genes (i.e., maternally expressed) were not expressed by Day-4 embryos (Cluster 1)…

Undoubtedly the network study depicted in Suppl Fig 3 represents the actual gene relationship, but the tangled mess is scarcely useful to identify something. The explanations in text given by the authors suffice.

Reviewer #2: Overall, the paper entitled "Maternal Metabolic Status and In-vitro Culture Conditions During Embryonic Genome Activation Deregulate the Expression of Energy-Related Genes in the Bovine 16-Cell Embryo" by Rabaglino et al. is well-written and effectively communicates the objectives, methods, and findings of the study. The paper demonstrates a high level of clarity and coherence in its writing. The language used is precise and accessible, allowing the reader to easily understand the research concepts and findings. The methodology section is thorough and well-described, allowing the reader to understand the experimental design and procedures employed in the study, especially regarding the model that was extensively characterized in previous works. The results section presents the findings of the study in a clear and concise manner, but I recommend that the authors increase the font size of the figures as they are currently difficult to read.

My main concern lies with the discussion section of the paper. The authors propose a model to examine the impact of metabolism on the molecular characteristics of embryos. However, there seems to be insufficient attention given to this aspect. In the results section, four main classes of genes were identified: those related to epigenetic mechanisms, OXPHOS, mTOR, and translation. Although the authors acknowledge their interconnectedness and susceptibility to environmental influences, the discussion section lacks reference to how the environment might be influencing these molecular statuses.

For example, the results indicate that embryos derived from LACT exhibited lower OXPHOS activity compared to DRY embryos, even in a supposed environment with low glucose and high free fatty acids (FFA). This raises questions about how to explain this observation. Besides, why did the LACT environment lead to embryos resembling those from in-vitro culture (IVC) based on the composition of both systems? How might these molecular changes in the environment be influencing the expression of genes related to epigenetics? These are some of the points that could bring valuable information to the discussion.

Given that the authors have already extensively characterized this model, I strongly suggest utilizing all available data to help elucidate these intriguing molecular findings. By incorporating additional information from their previous research, they can provide a more comprehensive explanation for the observed molecular changes and their relationship to the environment.

6. PLOS authors have the option to publish the peer review history of their article (what does this mean?). If published, this will include your full peer review and any attached files.

Reviewer #1: No

Reviewer #2: No

---

## [Author Response · Author response to Decision Letter 0]

28 Jul 2023

Although both reviewers found some interesting findings, they also have major concerns. Please respond to the reviewers' questions and submit the updated version of the manuscript. Particularly, the discussion should be made more precisely based on the findings of this study.

AU: We appreciate the positive feedback and the constructive criticism of the manuscript by both reviewers. We have addressed all of the comments by providing a response below and, where appropriate, by modifying the manuscript.

Reviewer #1: This study explores the impact of metabolic stress caused by milk production of dairy cows in the early embryo, in particular during the -major- embryonic genome activation (EGA). The transcriptome of embryos entirely cultured in vitro was examined in contrast with embryos fertilized in vitro that developed in the genital tract of dry and lactating cows. The analysis identified a number of (new) genes / pathways differentially expressed between all types of embryos used. The design of this study is elegant, the work is well written and easy to read.

AU: We appreciate the positive feedback and the constructive criticism of the manuscript.

The main limitations of this study are the following

- High dimensionality: in spite the correct limitative statistical tests for false discovery, authors should be cautious when drawing conclusions from this manuscript. Variability is also limited within in vivo embryos, as all them were collected from a same cow (L159-160). The above concerns must be discussed

AU: We agree with the reviewer on this point, and we have included this observation in the discussion (lines 383-388, clean manuscript). Still, the plot corresponding to a principal component analysis (a dimension reduction technique; FigS1) constructed with the whole transcriptome, which depicts clusters of samples based on their similarity, showed that samples from the same treatment group clustered close together. Thus, even though the sample size was small and variability was limited within the in vivo embryos, the observed results reflected the treatment effect for all the samples in the group. 

- Limited sample numbers: The use of N=3 groups per treatment in large data experiments is discouraged, as detection of outliers is more difficult and, in the case of strong divergences between one and the other two experimental groups, discarding one group ruins the experiment.

AU: The sample size of 3 per group is considered as the minimum for an inferential analysis of data derived from RNAseq (PMID: 26813401). We agree that is not possible to discard any sample if there are differences between them. Still, we are confident that the data generated with our experiment led us to the correct biological interpretation. Each sample was composed of a pool of 5 embryos, the results are biologically sound, and, as mentioned above, the principal component analysis plot shows that samples from each treatment group together when considering the whole transcriptome. In other words, there were not strong divergences between samples in the same treatment group. 

- Some inconsistencies of the Discussion section need re-thinking:

L386 onwards: the results dealing with epigenetic control (i.e., H2A, H3, and H4 and the methylation process, exhibited similar expression between IVP embryos at the 2- to 4-cell (pre-EGA) and 8 to 16-cell (post-EGA)) can be difficult to explain. The authors provide appropriate references to EGA activation (42 and 43), but the facts are not the same. In contrast, other results obtained in this study appear more consistent (L399-400)

AU: We agree with the reviewer. We have modified the paragraph to state that the transcriptomic results might indicate changes at the epigenome level, but we cannot prove it with this study (lines 396-400, clean manuscript).

L423-425: Or simply is an inconsistent result obtained from sample limitations.

AU: Perhaps. We added this possibility to the discussion (lines 427-428).

Other concerns

L39-41 (abstract): unclear sentence that should be re-rewritten.

AU: Thank you. This sentence was re-written.

L99-L112: Consider whether the text described is relevant for this study.

AU: Thank you. We believe that it is important to provide detailed information about the cows employed in this study, so the reader can have an overall idea of the experiment without needing to read the previous paper. 

L-244-245: re-write as follows: Functional analysis of DEG in each cluster revealed that 1413 Day 2 genes (i.e., maternally expressed) were not expressed by Day-4 embryos (Cluster 1)…

AU: Thank you, the sentence was rewritten.

Undoubtedly the network study depicted in Suppl Fig 3 represents the actual gene relationship, but the tangled mess is scarcely useful to identify something. The explanations in text given by the authors suffice.

AU: It is true that the network is tangled. Still, it provides the idea that genes in the three pathways are strongly connected and it also represents the location of the key genes in the networks.

Reviewer #2: Overall, the paper entitled "Maternal Metabolic Status and In-vitro Culture Conditions During Embryonic Genome Activation Deregulate the Expression of Energy-Related Genes in the Bovine 16-Cell Embryo" by Rabaglino et al. is well-written and effectively communicates the objectives, methods, and findings of the study. The paper demonstrates a high level of clarity and coherence in its writing. The language used is precise and accessible, allowing the reader to easily understand the research concepts and findings. The methodology section is thorough and well-described, allowing the reader to understand the experimental design and procedures employed in the study, especially regarding the model that was extensively characterized in previous works. The results section presents the findings of the study in a clear and concise manner, but I recommend that the authors increase the font size of the figures as they are currently difficult to read.

AU: Thank you very much for your positive feedback and constructive criticism. The figures were modified to facilitate the reading. 

My main concern lies with the discussion section of the paper. The authors propose a model to examine the impact of metabolism on the molecular characteristics of embryos. However, there seems to be insufficient attention given to this aspect. In the results section, four main classes of genes were identified: those related to epigenetic mechanisms, OXPHOS, mTOR, and translation. Although the authors acknowledge their interconnectedness and susceptibility to environmental influences, the discussion section lacks reference to how the environment might be influencing these molecular statuses.

For example, the results indicate that embryos derived from LACT exhibited lower OXPHOS activity compared to DRY embryos, even in a supposed environment with low glucose and high free fatty acids (FFA). This raises questions about how to explain this observation. Besides, why did the LACT environment lead to embryos resembling those from in-vitro culture (IVC) based on the composition of both systems? How might these molecular changes in the environment be influencing the expression of genes related to epigenetics? These are some of the points that could bring valuable information to the discussion.

Given that the authors have already extensively characterized this model, I strongly suggest utilizing all available data to help elucidate these intriguing molecular findings. By incorporating additional information from their previous research, they can provide a more comprehensive explanation for the observed molecular changes and their relationship to the environment.

AU: We appreciate this comment. We have thoroughly revised and modified the discussion to provide potential explanations to the observed results between LACT, DRY or IVP embryos (lines 438-470, clean manuscript)

---

## [Decision Letter · Decision Letter 1]

15 Aug 2023

Maternal metabolic status and in-vitro culture conditions during embryonic genome activation deregulate the expression of energy-related genes in the bovine 16-cells embryo

PONE-D-23-15571R1

Dear Dr. Lonergan,

We’re pleased to inform you that your manuscript has been judged scientifically suitable for publication and will be formally accepted for publication once it meets all outstanding technical requirements.

Kind regards,

Birendra Mishra, DVM, PhD

Academic Editor

PLOS ONE

Additional Editor Comments (optional):

Authors adequately responded to the reviewers comments. Thanks

Reviewers' comments:

Reviewer's Responses to Questions

**Comments to the Author**

1. If the authors have adequately addressed your comments raised in a previous round of review and you feel that this manuscript is now acceptable for publication, you may indicate that here to bypass the “Comments to the Author” section, enter your conflict of interest statement in the “Confidential to Editor” section, and submit your "Accept" recommendation.

Reviewer #1: All comments have been addressed

2. Is the manuscript technically sound, and do the data support the conclusions?

Reviewer #1: Yes

3. Has the statistical analysis been performed appropriately and rigorously? 

Reviewer #1: Yes

4. Have the authors made all data underlying the findings in their manuscript fully available?

Reviewer #1: Yes

5. Is the manuscript presented in an intelligible fashion and written in standard English?

Reviewer #1: Yes

6. Review Comments to the Author

Reviewer #1: In the opinion of this reviewer, the authors have successfully addressed all questions posed by the reviewers and the manuscript can be accepted for publishing.

7. PLOS authors have the option to publish the peer review history of their article (what does this mean?). If published, this will include your full peer review and any attached files.

Reviewer #1: No

---

## [Editor Report · Acceptance letter]

17 Aug 2023

PONE-D-23-15571R1 

Maternal metabolic status and in-vitro culture conditions during embryonic genome activation deregulate the expression of energy-related genes in the bovine 16-cells embryo 

Dear Dr. Lonergan:

I'm pleased to inform you that your manuscript has been deemed suitable for publication in PLOS ONE. Congratulations! Your manuscript is now with our production department. 

Kind regards, 

on behalf of

Dr. Birendra Mishra 

Academic Editor

PLOS ONE